# Mandrill mothers associate with infants who look like their own offspring using phenotype matching

Marie JE Charpentier[1]*[†], Clémence Poirotte[2†], Berta Roura-Torres[2,3], Paul Amblard-Rambert[3], Eric Willaume[4], Peter M Kappeler[2], François Rousset[1‡], Julien P Renoult[5‡]

[1]ISEM, Univ Montpellier, CNRS, IRD, EPHE, Montpellier, France; [2]Behavioral Ecology and Sociobiology Unit, German Primate Center, Leibniz Institute of Primate Research, Göttingen, Germany; [3]Projet Mandrillus, SODEPAL, Bakoumba, Gabon; [4]SODEPAL-COMILOG, Bakoumba, Gabon; [5]CEFE, Univ Montpellier, CNRS, EPHE, IRD, Montpellier, France

*For correspondence:
marie.charpentier@umontpellier.fr

[†]These authors contributed equally to this work
[‡]These authors also contributed equally to this work

Competing interest: The authors declare that no competing interests exist.

**Abstract** Behavioral discrimination of kin is a key process structuring social relationships in animals. In this study, we provide evidence for discrimination towards non-kin by third-parties through a mechanism of phenotype matching. In mandrills, we recently demonstrated increased facial resemblance among paternally related juvenile and adult females indicating adaptive opportunities for paternal kin recognition. Here, we hypothesize that mandrill mothers use offspring's facial resemblance with other infants to guide offspring's social opportunities towards similar-looking ones. Using deep learning for face recognition in 80 wild mandrill infants, we first show that infants sired by the same father resemble each other the most, independently of their age, sex or maternal origin, extending previous results to the youngest age class. Using long-term behavioral observations on association patterns, and controlling for matrilineal origin, maternal relatedness and infant age and sex, we then show, as predicted, that mothers are spatially closer to infants that resemble their own offspring more, and that this maternal behavior leads to similar-looking infants being spatially associated. We then discuss the different scenarios explaining this result, arguing that an adaptive maternal behavior is a likely explanation. In support of this mechanism and using theoretical modeling, we finally describe a plausible evolutionary process whereby mothers gain fitness benefits by promoting nepotism among paternally related infants. This mechanism, that we call 'second-order kin selection', may extend beyond mother-infant interactions and has the potential to explain cooperative behaviors among non-kin in other social species, including humans.

## Editor's evaluation

This study shows that, 60 years after the development of kin selection theory, new implications are still being uncovered. The authors report results of a long-term field study of a mandrill population in the forests of Gabon. Facial-pattern analyses and accompanying theoretical work support the hypothesis that females "socially engineer" relationships between their offspring and other offspring, based on facial resemblance. Via this mechanism, mothers appear to promote associations with individuals that are more likely to treat them as relatives, increasing the likelihood of future benefits from cooperative interactions. The authors suggest that their discovery could explain cooperative behaviours among non-kin in other social species, including humans.

## Introduction

Kin selection is an evolutionary process promoting traits that provide fitness benefits for genetic relatives of individuals expressing them (*Hamilton, 1964*). Empirical observations of diverse interactions arising from kin selection have been pervasively reported in natura and constitute the foundations of many studies on social evolution (*Clutton-Brock, 2002*). Kin selection sometimes necessitates kin recognition, which can operate through phenotype matching (*Penn and Frommen, 2010*). This mechanism, based on learning processes of, for example, odors, sounds, or visual cues, allows individuals to recognize kin based on phenotypic resemblance either with self ('self-referent phenotype matching' *Hauber and Sherman, 2001*; *Mateo and Johnston, 2003*) or with other kin used as templates (*Sherman et al., 1997*). Self-referent phenotype matching requires individuals to evaluate their own phenotype (e.g. through smell in rodents *Mateo and Johnston, 2003*; *Mateo, 2010*), which may be challenging in some situations. For example, although face recognition is a crucial prerequisite for visual communication and therefore for the maintenance of social relationships in many species (*Sheehan and Tibbetts, 2011*), including our own lineage (e.g. *Alvergne et al., 2009*; *DeBruine et al., 2008*; *Parr, 2011*; *Sheehan and Nachman, 2014*), in natural contexts, animals other than humans have probably limited access to cues regarding their own facial traits (but see *Hauber and Sherman, 2001*; *Hauser et al., 1995*). Using familiar kin as templates to recognize unfamiliar kin also requires particular conditions, including the presence of relatives during template formation. This mechanism has been rarely demonstrated in the wild (but see *Levréro et al., 2015* and *Holmes, 1986*; *Mateo, 2003* for lab studies). A crucial question is therefore how an individual may recognize unfamiliar kin when it cannot match phenotypes to itself or to other templates.

Mandrills are non-human primates from Central Africa that live in large matrilineal societies, characterized by family units of philopatric, maternally-related and highly nepotistic females. Males, the dispersing sex, are only temporary residents in these groups, and the highest-ranking male sires a large proportion of infants born into different matrilines each year (*Charpentier et al., 2005*; *Charpentier et al., 2020*). Each cohort of new-borns therefore includes numerous paternal half-sibs that behaviorally discriminate each other from non-kin. Indeed, both maternal and paternal half-sibs associate, groom, but also aggress each other more compared to non-kin, as early as juvenility (*Charpentier et al., 2007*), and this kin bias extends until adulthood, at least among the philopatric female mandrills (*Charpentier et al., 2020*). Previously, we have further demonstrated that unfamiliar kin recognize each other using phenotype matching based on acoustic (*Levréro et al., 2015*) and possibly visual cues (*Charpentier et al., 2017*), and that facial resemblance correlates positively with genetic relatedness across female mandrills (*Charpentier et al., 2020*).

Crucially, paternal half-sisters resemble each other visually more than maternal half-sisters do (*Charpentier et al., 2020*), even though both kin categories share, on average, similar degrees of genetic relatedness ($r\sim0.25$; and see Appendix 1). This heightened facial resemblance among paternal half-sisters compared to maternal half-sisters, possibly resulting from genomic imprinting processes (*Haig, 2002*; *Moore and Haig, 1991*), indicates adaptive opportunities for paternal kin recognition, necessarily mediated by phenotype matching mechanisms. Using self-referent phenotype matching to discriminate paternal half-sibs appears, however, unrealistic in wild mandrills because of environmental constraints (no physical medium to allow facial self-recognition). Facial similarity could also result from selection processes on other self-evaluable phenotypic traits such as body odors, but this mechanism appears less parsimonious to explain the *increased* facial resemblance observed among paternal half-sisters (Appendix 1) than a mechanism based on direct selection on facial traits to provide cues of paternity. Using the father's face as a template is also unlikely in mandrills because of the highly pronounced sexual dimorphism and morphological differences between adults and immatures in this species (infants do not resemble their father: *Appendix 1—figure 1* and *Appendix 1—table 1*).

In (*Charpentier et al., 2020*), we discussed an alternative mechanism where third-parties would use increased facial resemblance among paternal half-sibs to shape their social relationships. We proposed that mothers could evaluate their offspring's facial resemblance with other youngsters and guide their offspring's social opportunities towards similar-looking ones, paving the way for the differentiated social relationships that we previously reported among paternal half-sibs. If mothers indeed manipulate their offspring's social preferences, we expect this to occur very early during development, in infants aged ≤1 yr, for two reasons. First, in nonhuman primates, the first year of life represents a developmental stage where infants are still under strong maternal control (*Maestripieri, 2018*).

Second, the average difference in facial resemblance among paternal half-sisters compared to maternal half-sisters or non-kin was higher during juvenility (2–4 yrs) than during adulthood (*Charpentier et al., 2020*), suggesting that cues of paternity are present, and particularly pronounced, early in life.

In this study, we provide empirical evidence that mothers and their offspring both associate more with similar-looking other infants, who are more likely than expected by chance to be their offsprings' paternal half-sibs. However, because paternal kin only share paternal genes, this maternal behavior cannot be explained as a standard form of kin selection, which requires relatedness between an actor (here the mother) and a non-descendant recipient (here the similar-looking infant). To address the question of how and whether such a behavior may evolve, we formally demonstrate an evolutionary mechanism by which mothers may gain fitness benefits from favoring nepotism between their own offspring and their offsprings' paternal half-sibs, as infants themselves benefit from interacting repeatedly with kin, following predictions of the kin selection theory. The mechanism that we propose to call 'second-order kin selection', can be generalized beyond mother-infant interactions. While it is a novel explanation for the evolution of nepotism, second-order kin selection perfectly fits into the mathematical framework offered by inclusive fitness theory (*Gardner et al., 2011*; *Hamilton, 1964*).

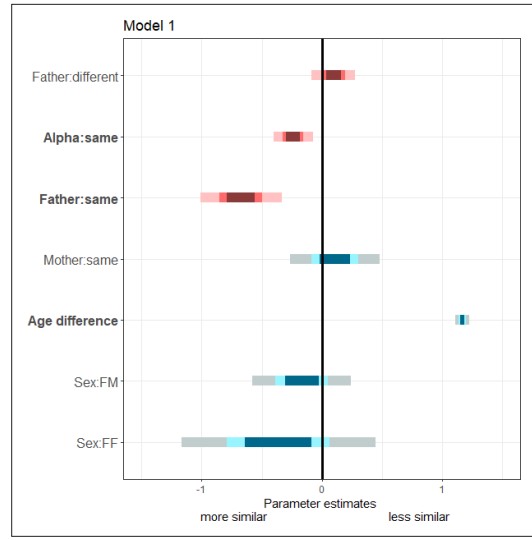

**Figure 1.** Summary of fixed effects' parameters included in the model analyzing average facial distance among infants. For each estimate, the 50% (inner), 70% (middle), and 95% (outer) Wald confidence intervals are shown. Pink shades highlight variables of interest while blue shades correspond to control variables. The following variables are displayed: father identity (top three rows: infants born to different fathers -'Father:different'-; infants conceived during the same alpha male tenure -'Alpha:same'-; infants born to the same father -'Father:same'-; reference category: Alpha:different); mother identity (infants born to the same mother -'Mother:same'-; reference category: Mother:different); infants' difference in age ('Age difference'); infants' difference in sexes ('Sex:FM' and 'Sex:FF'; reference category: MM). Bold y-axis labels highlight variables with significant effects (p<0.05).

## Results and discussion
### Empirical evidence in mandrills
#### Increased resemblance among paternal half-sibs during infancy
We first show that, during infancy, paternal half-sibs resemble each other more than maternal half-sibs and non-kin do. We retrained VGG-Face, a popular algorithm for human face recognition, to recognize 112 individual mandrills from their face, independently of their position, lighting or facial expression, using approximately 12 k training pictures. These pictures were taken in the course of a long-term field project on a large, natural social group of mandrills in Gabon. We used this retrained model to predict facial distance (calculated within the encoding space of the model, see Materials and methods) among 80 individually recognized infants (0–1 yr) of both sexes (39 females, 41 males), represented by a total of 204 new pictures (1–10 pictures/infant), not used for training the model. Maternal identity was known for all 80 infants based on direct observations. Due to field constraints, paternal identity was determined genetically for only 32 infants. For the remaining 48 infants with unknown paternity, we differentiated between pairs of infants conceived during the tenure of the same alpha male from those conceived under tenures of different alpha males. Indeed, reproductive skew in mandrills is high, with alpha males monopolizing 60–70% of all conceptions (*Charpentier et al., 2005*; *Charpentier et al., 2020*). While infants born during the tenures of 'different alpha males' are thus most likely not paternally related, those born during the tenure of the 'same alpha male' presumably include a large proportion of paternal half-sibs. Using a linear mixed model (LMM) on this large dataset (N=2556 dyads), and controlling for infant age and sex as well as maternal identity, we found

**Table 1.** Predictors for facial distance among infants.
Significant predictors (p<0.05) are in bold (LMM with a Gaussian response).

| | | Infants' facial distance N=2556 dyads | | | |
|---|---|---|---|---|---|
| | | $\chi^2$ | $p > \chi^2$ | Estimate | Standard error |
| Predictors | Father/alpha male at infants' conception* | 28.07 | <0.001 | Same alpha: –0.24 Different father: 0.10 Same father: –0.67 | 0.08 0.09 0.17 |
| | *Same alpha vs. same father* | 5.87 | **0.015** | | |
| | *Different alphas vs. different fathers* | 1.06 | 0.30 | | |
| | *Same alpha vs. different alphas* | 8.16 | **0.004** | | |
| | Infants' mother(s)† | 0.32 | 0.57 | 0.12 | 0.19 |
| | Infants' difference in age | 1231.8 | <0.001 | 1.18 | 0.03 |
| | Infants' sex‡ | 0.84 | 0.66 | FF: –0.36 | 0.41 |
| | | | | FM: –0.17 | 0.21 |

*Reference: different alpha males.
†Reference: different mothers.
‡Reference: MM (male-male).

that paternal half-sibs resemble each other facially more than any other dyads of infants. In addition, these paternal half-sibs, but also those conceived during the tenure of the same alpha male, resemble each other significantly more (lower average facial distance; *Figure 1*, *Table 1*) than those sired by two different fathers or conceived during the tenures of two different alpha males (higher average facial distance). In contrast, at these young ages, maternal half-sibs do not look more alike than infants born to different mothers (*Table 1*). This result therefore strongly supports and extends one of our previous key results: starting as early as infancy (this study) and continuing throughout juvenility and adulthood for females (*Charpentier et al., 2020*), paternal half-sibs resemble each other more than maternal half-sibs do.

## Mothers and their offspring associate more with similar-looking other infants

We then restricted our image data set to those infant dyads for which we had detailed records on spatial associations routinely collected during behavioral observations (N=48 infants and their 30 mothers). Using generalized linear mixed models (GLMM with negative binomial family for over-dispersed count data), we analyzed spatial associations (i) among infants ('infant-infant', N=282 dyads); (ii) between infants and other mothers ('mother-infant', N=580); and (iii) among mothers having infants at the same time ('mother-mother', N=325) as a function of the residuals (see Materials and methods) of facial distance among infants, and controlling for matrilineal origin, maternal relatedness and rank, and infant sex and age (at the time of observations).

We found that mothers associate significantly more with infants that look like their own infants compared to infants that do not (*Figure 2*, *Table 2*). Infants also associate significantly more with other infants that look more like them. In contrast, associations among mothers are not influenced by the average facial distance of their offspring (*Figure 2*, *Table 2*), although both mothers should be also spatially close (but more distant) given mother-offspring association patterns (*Appendix 1—figure 2*). This last result suggests that mother-infant and infant-infant association patterns do not emerge as by-products of mother-mother association patterns. Pre-existing friendships among mothers, or among these mothers and a common father (*Appendix 1—table 1*), are thus unlikely to explain the relationships observed between mother-infant and infant-infant dyads. This interpretation is further supported by the analysis of grooming relationships among mothers having infants at the same time ('mother-mother', N=310 dyads) as a function of the residuals of facial distance among infants, and the same confounding variables as above (Materials and methods). As for spatial associations, grooming relationships among mothers are not related to facial resemblance of their offspring (LRT, $\chi^2$=1.71, p=0.19). Similar models on infant-infant and mother-infant grooming relationships were not performed because infants never groom each other, and, of a total of 560 mother-infant dyads, only 6

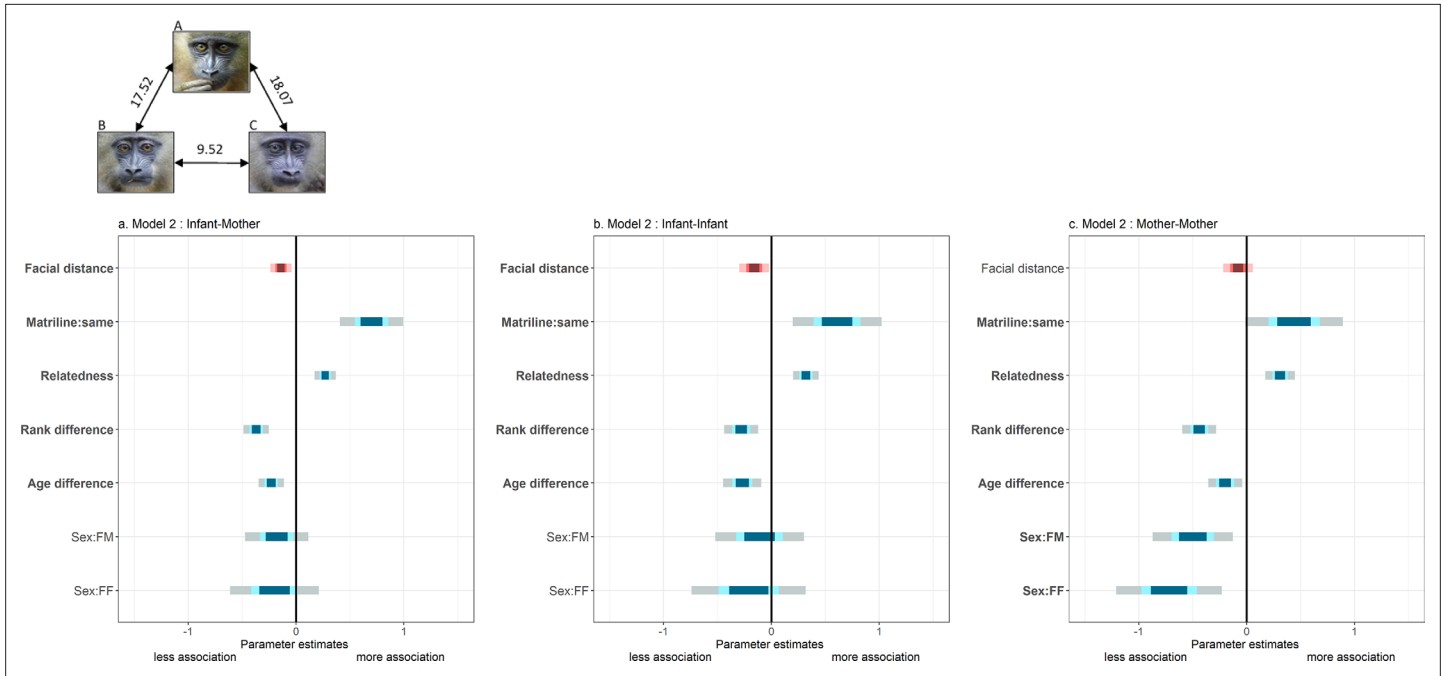

**Figure 2.** Summary of fixed effects' parameters included in the three models analyzing spatial association across dyads of (**a**) mothers and other infants; (**b**) infants; and (**c**) mothers. For each estimate, the 50% (inner), 70% (middle) and 95% (outer) Wald confidence intervals are shown. Pink shades highlight the variable of interest ('facial distance') while blue shades correspond to control variables. Note that a negative estimate (as for 'facial distance') indicates a negative correlation between spatial association and that variable ('facial distance'): individuals associate more with low values of 'facial distance' (high resemblance). The following variables are displayed: infants' residuals of facial distance ('Facial distance'); different vs. same mothers' matrilines ('Matriline:same'; reference category: different); mothers' relatedness ('Relatedness'); mothers' difference in rank ('Rank difference'); infants' difference in age ('Age difference'); infants' difference in sexes ('Sex:FM' and 'Sex:FF'; reference category: MM). Bold y-axis labels indicate variables with significant effects (p<0.05). Pictures depict three male infants with their average facial distances: B and C resemble each other most, in contrast to A.

were observed grooming each other during the course of this study. An important implication of this last observation is that mothers probably bring infants into proximity to each other but do not invest into other social behaviors such as grooming (i.e. infants 'do the rest').

In mandrills, mothers constitute the main social partner for their offspring for several months or years. During their first year of life, infants are closely associated with their mother (***Appendix 1—figure***

**Table 2.** Predictors of the spatial associations recorded across mother-infant, infant-infant and mother-mother dyads. Significant predictors (p<0.05) are in bold (GLMM with negative binomial response family and log link). The reported dispersion parameter is the so-called shape parameter of the negative binomial distribution.

|  |  | Mother-infant association N=580 dyads Dispersion parameter = 2.43 | | | | Infant-infant association N=282 dyads Dispersion parameter = 2.635 | | | | Mother-mother association N=325 dyads Dispersion parameter = 1.281 | | | |
|---|---|---|---|---|---|---|---|---|---|---|---|---|---|
|  |  | $\chi^2$ | $p > \chi^2$ | Estimate | Standard error | $\chi^2$ | $p > \chi^2$ | Estimate | Standard error | $\chi^2$ | $p > \chi^2$ | Estimate | Standard error |
| Predictors | Infants' facial distance (residuals) | 7.72 | **0.005** | –0.14 | 0.05 | 5.30 | **0.021** | –0.16 | 0.07 | 1.33 | 0.248 | –0.08 | 0.07 |
|  | Infants' matriline* | 18.11 | **<0.001** | 0.70 | 0.15 | 7.85 | **0.005** | 0.61 | 0.21 | 3.45 | 0.063 | 0.44 | 0.23 |
|  | Mothers' relatedness | 31.17 | **<0.001** | 0.27 | 0.05 | 24.04 | **<0.001** | 0.32 | 0.06 | 17.01 | **<0.001** | 0.31 | 0.07 |
|  | Mothers' rank difference | 37.47 | **<0.001** | –0.37 | 0.06 | 10.74 | **0.001** | –0.28 | 0.08 | 28.49 | **<0.001** | –0.44 | 0.08 |
|  | Infants' age difference† | 13.77 | **<0.001** | –0.23 | 0.06 | 10.49 | **0.001** | –0.27 | 0.09 | 6.20 | **0.013** | –0.20 | 0.08 |
|  | Infants'sex† | 1.39 | 0.499 | FF: –0.20 FM: –0.18 | FF: 0.21 FM: 0.15 | 0.66 | 0.719 | FF: –0.21 FM: –0.11 | FF: 0.27 FM 0.21 | 8.94 | **0.011** | FF: –0.72 FM: –0.50 | FF: 0.25 FM: 0.19 |

*Reference: different matrilines.
†Reference: MM (male-male).

*2*). To confirm that mothers, and not infants, drive patterns of association with similarly-looking other infants, we analyzed the directionality of mother-offspring follows and approaches. We found that infants follow and approach their mothers in 97.6% and 89.4% of these events, respectively (follow: N=2,387 total events recorded on 139 mother-infant dyads; approach: N=4,482 total events recorded on 155 mother-infant dyads). Therefore, in the vast majority of cases, infants go, indeed, where their mothers are or go. Finally, we studied the correlation between the rate of association recorded among the 282 dyads of infants and the average rate of association recorded between each of these infants and the mother of the other infant (*Appendix 1—figure 3*): 71% of the variance in infant-infant associations is explained by the average mother-infant association (and vice-versa; Pearson correlation test: *r*=0.85, p<0.0001). Altogether, these correlational results support the hypothesis that mothers are responsible for the increased association observed among infants that depends on several factors, including infants' facial resemblance.

In other mammals, from rodents to humans, mothers often show some form of control over social opportunities and preferences of their offspring, and maternal behavior may impact both their infants' neural and social development (*Rilling and Young, 2014*). In chimpanzees, mothers with sons are more gregarious and associate more with adult males than mothers with daughters, thereby shaping their sons' social trajectories (*Murray et al., 2014*). Despite strong maternal control over offspring's social choices in mammals (*East et al., 2009*; *Rilling and Young, 2014*), we explore below an alternative scenario where fathers, not mothers, may use prominent visual resemblance among paternal half-sibs to target paternal care and mediate nepotism among them, thereby ultimately favoring the transmission of paternal genes.

## Alternative hypothesis: Is increased resemblance among paternal half-sibs driven by father-based selection?

To care for their own offspring, mandrill fathers need some form of paternity certainty because females mate with several males around ovulation (*Setchell, 2016*). In some other promiscuous species, true paternal care has been unambiguously demonstrated: male baboons, for example, selectively support their own offspring in agonistic disputes (*Buchan et al., 2003*). If mandrill fathers know with certainty at least one or a few of their offspring, then increased facial resemblance may help discriminating others. This mechanism, although plausible, is subject to cognitive constraints and a precise knowledge about female fertility as female mandrills are only fertile during a restricted time window (May-Sept *Dezeure et al., 2022*). More parsimoniously, mothers may remember whom they mated with and associate with the putative father after birth to promote paternal care. Heightened facial resemblance among paternal half-sibs could then help fathers to discriminate all of their offspring from other infants. Contrary to male baboons, however, paternal care in mandrills is elusive: males neither carry infants nor do they groom or affiliate with them (MJEC pers. obs.), even though in captivity, males are spatially closer to their own juvenile offspring than to unrelated juveniles of similar age (*Charpentier et al., 2007*). While we cannot exclude paternal care as a driving force for increased facial resemblance among paternal half-sibs, patterns of male residency in this natural mandrill population do not provide strong support for this scenario. Indeed, males are mainly present during the short breeding season, although some males remain in the group from a few days to a few months (*Brockmeyer et al., 2015*). Among 69 male.years (N=29 different subordinate and dominant males) observed throughout 8 reproductive seasons, only 44.7% and 54.5% of males, respectively, that were present during a given reproductive season remained in the study group until the next birth season. Opportunities for fathers to personally care for their infants therefore occur statistically only half of the time. Finally, those males that were present during a given reproductive cycle are probably not responsible for maintaining proximity with either infants or their mothers (MJEC and BRT, pers. obs.), but exploring this alternative mechanism would require detailed data on male-infant relationships that are not available yet. To conclude, we present empirical evidence that mandrill mothers may influence offsprings' association patterns towards similar-looking infants, and our knowledge of the natural history of mandrills makes less plausible fathers to be also involved in this process, although we are currently unable to analytically exclude this alternative mechanism with available data. In the future, showing that fathers rather than non-fathers are more likely to stay in the group at their offsprings' birth and that they actively intervene in their offsprings' social interactions would help to disentangle the respective role of mothers and fathers in offsprings' social development.

## The mother's benefits of matching her infants with related ones

Because paternal half-siblings share paternal, not maternal genes, we finally show how, under minimal assumptions, mothers can obtain fitness benefits by fostering interactions between their offspring and paternally-related siblings. Key steps to that conclusion are that infants engage in repeated interactions within dyads and that such interactions are mostly of two types, 'repeatedly cooperate' or 'repeatedly defect', so that only dyads of cooperators can repeatedly cooperate. Then, any maternal behavior that increases the likelihood that her offspring will repeatedly cooperate rather than repeatedly defect may be favored. This two-step reasoning does not require a statistical association (linkage disequilibrium) between alleles affecting the control of assortment of infants by mothers, and alleles affecting cooperation. To reach this conclusions, we will use a 'one-generation' formalism (*Lehmann and Rousset, 2014*) suitable to take into account both the interactions between relatives and multilocus processes, and which has proven useful in particular to avoid double counting of fitness benefits (as may happen when compounding fitness effects of related individuals over different generations). In the present case, it allows to correctly account for fitness benefits to mothers when infants are involved in social interactions.

In this formalism, selection on an allele affecting the mother's fitness $W$ is quantified as a covariance (*Price, 1970*) between an indicator variable $z_f$ for presence of the allele in a mother, and mother's fitness. More specifically the expected effect of selection on the change in frequency of the allele over one generation is proportional to $\text{cov}(z_f, W)$ (it would be equal to this covariance if all adults expressed the allele, rather than only females). $W$ is her expected number of (adult) offspring, expressed as a function $w(x_f, x_p)$ of her own behavior $x_f$ and of that $x_p$ of her social partners (neighbor-modulated fitness), and then as a function of gene effects underlying such behaviors (that is, the value of $x_f$ and $x_p$ are conceived as functions of the allele borne by the respective individuals, e.g. $x_f = x_0 + z_f\delta$ may represent the extent of female associative behavior whether she bears ($x_f = x_0 + \delta$ when $z_f = 1$) or not ($x_f = x_0$ when $z_f = 0$) the allele under consideration). The covariance expression for selection then involves expected values of products of the mother's and partners' indicator variables, that can be interpreted in terms of relatedness coefficients, and linkage disequilibria when several loci are involved. In the present case, we will see that many such products can be ignored, allowing a simple characterization of selection on maternal behavior.

For simplicity, we assume effects on infant survival. We could alternatively assume effects on infant reproductive potential ('quality') but then the fitness of a mother should be measured as a weighted sum of numbers of offspring of different quality, which would complicate the model formulation without modifying its conclusions. Likewise, formal models including pairing processes are generally complex to formulate but such a formulation is not required to understand the key qualitative features of the present scenario.

First, consider the fitness benefits for paired infants. Let us assume that pairs of infants play an iterated prisoner's dilemma in its canonical form (with $R$ indicating the 'rewards' received by two cooperating infants, $P$ the 'punishment' payoff received by two defecting infants, $T$ the 'temptation' payoff received by a defecting infant interacting with a cooperating infant, and $S$ the 'sucker's' payoff received by a cooperating infant interacting with a defecting infant, and $w$ the probability of iteration [*Axelrod and Hamilton, 1981*]), and a tit-for-tat strategy in response to defection. Accordingly, after multiple interactions the expected payoffs are $R/(1-w)$ for pairs of cooperators, $P/(1-w)$ for pairs of defectors, and respectively $S + Pw/(1-w)$ and $T + Pw/(1-w)$ for a cooperator and a defector paired together. Provisionally assuming that cooperators and defectors are equally frequent, the average payoff is therefore $(R+P)/(2(1-w))$ for identical dyads, and $((1-w)(S+T) + 2Pw)/(2(1-w))$ for non-identical dyads. For long-lasting interactions ($w \to 1$), the relative values of the payoffs of identical and non-identical dyads compare essentially as $R + P$ vs. $2P$. Given that cooperation is mutually beneficial ($R > P$), identical dyads are favored over non-identical ones. If cooperators and defectors are not equally frequent, the average payoff of identical dyads will scale as a weighted average of $R$ and $P$ but the reasoning and main conclusion is otherwise unchanged: identical dyads still enjoy an average fitness benefit proportional to $R - P$. This conclusion embodies the fact that, on average over cases where they cooperate and cases where they do not, individuals can benefit from increasing the likelihood of interacting with identically-behaved ones, whether relatives of not. Therefore, they can benefit by increasing the likelihood of interactions between relatives, and it would readily explain kin recognition by infants, which we do not assume here (see *Appendix 1—table 1*).

Next, consider selective effects on alleles acting in mothers to control the assortment of their infants. For simplicity, let us assume that payoffs affect infant survival, so that the mother's fitness is proportional to the survival probability of her offspring. We can write the expected payoff (and, up to a constant factor, the linearized survival benefits) for a focal infant as $W := \beta_0 + \beta_z z + \beta_{z_p} z_p + \beta_{zz_p} zz_p$, where $z$ is the indicator variable for the event that the focal infant initiates the interaction by cooperating, $z_p$ is the same variable for its partner, and the βs are functions of the model parameters. Then, if a 'mutant' allele increases by $\delta$, relative to a 'resident' allele, the likelihood that infants assort in identical pairs, this mutant will experience a total fitness effect proportional to $\delta \Delta W$, where $\Delta W := \beta_z \Delta E(z) + \beta_{z_p} \Delta E(z_p) + \beta_{zz_p} \Delta E(zz_p)$ where for any variable $x$, $\Delta E(x)$ is the difference in expected value of $x$ between infants of mothers bearing the mutant vs. those of mothers bearing the resident allele.

We first focus on the last term of the selective effect $\Delta W$, $\beta_{zz_p} \Delta E(zz_p)$ as we will later see that other terms are comparatively negligible. This represents the fitness effect of the interaction of events represented by $z$ and $z_p$. The system of four equations, implied by the expression for $W$ for all combinations of the two indicator variables, shows that $\beta_{zz_p}$ is the difference between the unweighted average payoff of identical dyads (z and $z_p$ equal to each other) and the unweighted average payoff of non-identical dyads. As shown above, this difference is proportional to $R - P > 0$. Consider further that phenotypes are affected by the additive effect of the alleles transmitted by the parents, $z = C_m + C_f$ where $C_m$ and $C_f$ are effects inherited from mother and father, respectively. Likewise, for the infant partner, $z_p = C_{m(p)} + C_{f(p)}$ in terms of effects $C_{m(p)}$ and $C_{f(p)}$ inherited from the partner's mother and father. Then the survival of an offspring will increase with any of the products $C_m C_{m(p)}$, $C_m C_{f(p)}$, $C_f C_{m(p)}$, and $C_f C_{f(p)}$.

We do not assume any relatedness among mothers (which would increase the expected value of $C_m C_{m(p)}$), nor do we assume any form of (dis-)assortative mating increasing or reducing the expected value of cross-sex products. Then, only $\delta \beta_{zz_p} \Delta E(C_f C_{f(p)})$ remains, meaning that any maternal behavior that increases the expected value of $C_f C_{f(p)}$ over offspring would enjoy fitness benefits. The conditions under which such benefits may outweigh costs possibly associated with the choice process are general conditions favoring choice: a high variance in quality (here, a high variance of inherited effects on cooperative behavior), and a strong impact of choice on expectation of $C_f C_{f(p)}$, which is dependent on a high variance in male reproductive success within a cohort of infants (the case in mandrills *Charpentier et al., 2005*; *Setchell et al., 2005*) and on the existence of cues to infer paternal relatedness between infants, here increased facial resemblance among paternal half-sibs.

This fitness effect results from the fact that infants being more similar at loci controlling phenotypic similarity are more similar at loci controlling cooperation. Such shared similarity ('identity disequilibrium') automatically results from an increased likelihood of shared paternity. It is expected, indeed, in any population where there is variation in relatedness among individuals, even in a finite panmictic population (*Sved, 1971*). When some pairs of individuals are more related to each other than other pairs, the fact that a pair is identical at one locus is an indication that the pair is more related than a random pair on average and then, it also tends to be more identical than random pairs at other loci in the genome (*Grafen, 1990*). By contrast, $\delta \Delta E(z)$ may be positive only if the alleles affecting the control of assortment of infants become statistically associated to cooperator alleles in mothers' genotypes. Such linkage disequilibria are typically of lower magnitude than identity disequilibrium, as recombination is generally efficient in reducing them (*Rousset and Roze, 2007*). Likewise, $\delta \Delta E(z_p)$ depends on a statistical association between the alleles affecting the control of assortment and cooperator alleles borne by the infant partner, and given that we do not assume that mothers are able to recognize any cooperator allele in any infant, such an association is all the more likely to be weak.

Although this mechanism does not assume any increased relatedness among mothers of interacting dyads over mothers of non-interacting dyads, such variation in relatedness may be difficult to fully exclude in natural populations. If it is present, a selection effect $\delta \beta_{zz_p} \Delta E(C_m C_{m(p)})$, analogous to the above one, arises and represents a standard kin selection effect since actor and recipient are then related. Yet, this kin-selection effect arises only in conditions where the previous effect arises and will be comparatively small when maternal relatedness is small relative to the relatedness between paternally-derived gene copies. Thus, even if present, a kin-selection effect will be small compared to the main selection effect favoring assortment between paternal half-sibs.

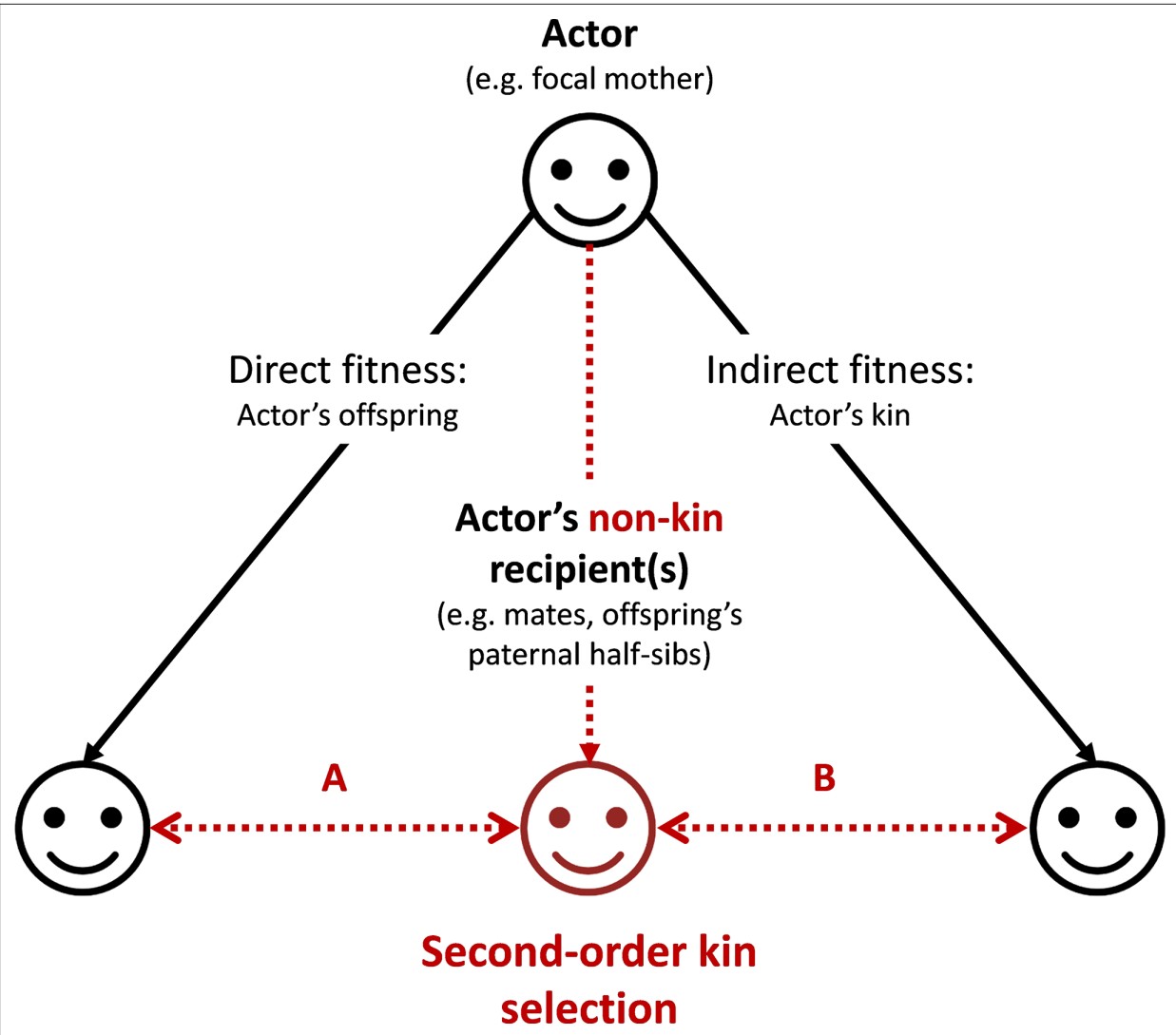

**Figure 3.** Graphical representation of the 'second-order kin selection' process. The actor (here a focal mother, face on the top) has the control over her behaviors with social partners (all three faces on the bottom including actor's offspring, other kin and non-kin in red). Plain black arrows represent the inclusive fitness framework of the kin selection theory. Dashed red arrows represent the second-order kin selection process where an actor's social behaviors towards a non-kin recipient (e.g. an offspring's paternal half-sib) favor the latter's social behaviors towards the actor's kin (offspring or other kin). We provide a few examples (A: the mechanism explored in this study; B: a generalization of the mechanism to other actor's kin) where this process may occur (and see discussion). Importantly, the second-order kin selection necessitates that non-kin recipient in red shares genetic or reproductive interests with the actor's kin (double-arrows).

The main conclusion is therefore that selection on mother's control of assortment of infants is proportional to $R - P$ and to the identity disequilibrium she creates by such control. By favoring nepotism between their own offspring and their paternal half-sibs, mothers would therefore derive direct fitness benefits (*Figure 3*, scenario A). The mechanism described here can be generalized to any actor whose behaviors promote positive interactions between a non-kin recipient and any actor's kin (*Figure 3*, scenario B). We term the mechanisms described in this generalized version 'second-order kin selection'.

## Conclusion and future directions

Given mandrills' biology, and based on the analytical evidence shown here, we propose that the most plausible interpretation of the heightened facial resemblance observed among paternally related individuals, from infancy to adulthood (*Charpentier et al., 2020*), is that this resemblance has been selected to allow mothers to foster interactions between their offspring and their offspring's

paternal relatives. The alternative, but not mutually exclusive explanation that fathers could reinforce this pattern cannot be totally excluded with current data. We propose the term 'second-order kin selection' to explain such maternal behavior. This process is, indeed, distinct from classic kin selection as actors (mothers) may derive direct fitness benefits from interacting with non-kin recipients (their offspring's paternal half-sibs) because of the benefits of these interactions towards the actors' own kin that occur through the increased expected payoffs of reciprocal interactions between kin.

Two critical conditions of this process are the ability to predict relatedness between assorted individuals (here by phenotype matching) and the occurrence of reciprocal interactions between the matched social partners. We provided evidence that the first condition is met in mandrills. In particular, we have both shown that cues of relatedness are present in mandrills' face at all ages and that these cues are likely used by mothers (although we do not assume this process to be conscious) who modify their behavior (here, spatial association) accordingly. A firm demonstration of the second condition, in particular that early associated individuals remain associated throughout their life and reciprocate more with each other than non-associated individuals, would necessitate longitudinal behavioral and demographic data that are not yet available for mandrills (but for preliminary results on patterns of association among 1- to 2-year-old juveniles, see *Appendix 1—figure 4*). However, reciprocal interactions among kin have been well demonstrated in other species. Females baboons, for example, form the most enduring and reciprocal social bonds with their close kin (*Silk et al., 2006*). In addition, the positive and strong relationship between individual fitness and the quality of the social environment, often approximated using measures of spatial association, is no longer in question. Indeed, social integration is one of the strongest predictors of health and longevity in humans (*Holt-Lunstad et al., 2010*; *Stringhini et al., 2017*). This conspicuous relationship is also observed in many social mammals – with effect sizes of strikingly high amplitudes (*Snyder-Mackler et al., 2020*). In all mammalian orders, individuals that are poorly socially integrated display considerably shorter life spans and decreased reproductive success than individuals enjoying a rich social life (*Silk et al., 2010*; *Barocas et al., 2011*; *Stanton et al., 2012*; *Archie et al., 2014*; *Vander Wal et al., 2015*). In semi-free ranging mandrills, notably, affiliation and grooming exchanged during juvenility translate into reproductive benefits in females in adulthood: females who are more socially integrated give birth on average a year before females that are less socially integrated (*Charpentier et al., 2012*).

To the best of our knowledge, the second-order kin selection as presented here, has been overlooked so far, and how this process shapes the social structure of animal societies is still unknown. Yet, cooperation among non-genetically related in-laws (affinal kin), often reported in humans (*Alvard, 2009*; *Koster, 2018*; *Salali et al., 2016*), fits into this selection model because affinal kin or spouses may share genetic interests in the next generation (see also *Burton-Chellew and Dunbar, 2011*; *Dyble et al., 2018*). In nonhuman animals, the generality of this process will depend on the taxonomic distribution of reciprocal interactions. Other modulating factors involve the variance in male parenting, which influences the benefits for a kin recognition mechanism, the conditions that the matched individuals are unable to perform this recognition themselves, and the opportunity for control of assortment of relatives. Several of these conditions may be widespread in animal societies. Although the opportunity for control of kin assortment may be the most restrictive condition, mothers guide their offspring's social development in many social mammals (*Maestripieri, 2018*; *Rilling and Young, 2014*) and other conditions similarly leading to direct selection for assortment of related offspring could operate elsewhere, such as in the various forms of cooperative breeding or nest parasitism in birds. Finally, our study highlights how state-of-the-art artificial intelligence algorithms combined with long-term field data can help unravelling adaptive details of social complexity that would otherwise go completely unnoticed.

## Materials and methods
### Study population
Since 2012 and within the framework of the 'Mandrillus Project' (*Brockmeyer et al., 2015*; *Charpentier et al., 2020*), we have been monitoring the only habituated, free-ranging and non-provisioned group of mandrills, living in a protected area and its surroundings (Lékédi Park, Bakoumba), in Southern Gabon. This group originated from 65 individuals that initially lived in a semi-free ranging population housed at CIRMF (Centre International de Recherches Médicales de Franceville, Gabon) and released

on two occasions into the protected area (2002 and 2006 *Peignot et al., 2008*). Captive-born females reproduced with wild immigrant males starting the first year post-release. In 2020, the group was composed of ca. 250 individuals of both sexes and all ages, about 200 of them being individually known and monitored on a daily basis; the remaining 7 captive-born adult females represented less than 3% of the animals studied for this project.

During daily observations, we record detailed data on group composition and individual behaviors. In addition, we take pictures of unambiguously identified individuals when visible and close (less than 10 m away). In this study, we considered a total of 80 infants (39 females, 41 males; aged 4–365 days) who contributed to the dataset of portrait images and 48 infants (24 females, 24 males) born to 30 different mothers contributing to the study of spatial associations as a function of facial resemblance (excluding those infants that were aged more than a year apart or those for whom no behavioral data was available). Dates of birth were exactly known for 31 infants and estimated within a time-window of one day to two months for the remaining 49 infants based on patterns of mother's sexual swellings and infants' physical appearance.

Although the 80 studied infants were born into the same natural social group, under similar environmental conditions and were chosen randomly, they constitute a sample of all infants born (N = 292 from 2012 to 2021). We cannot exclude this sample to be biased towards e.g. fearless individuals (that would be easier to observe than shy animals). However, we do not see how this possible sampling bias could have affected our results in a non-random direction as infants were sampled independently of their maternal or paternal origins.

## Measuring facial resemblance
### Image database and pre-processing

The mandrill face database includes ~16 k images representing 276 different mandrills originating from the study natural population (12.9 k images), a semi-captive population housed at CIRMF (Centre International de Recherches Médicales de Franceville, Gabon; 2.7 k images) and other sources (the Internet, the Wildlife Reserves of Singapore, Zoo of Grandy; 0.4 k images). Images from Gabon (natural and CIRMF populations) were taken between 2012 and 2018. Pictures represent individuals that are awake and passive, awake and active (i.e. feeding, grooming, vocalizing) or anesthetized during several captures that occurred between 2012 and 2015 (representing 1.1 k images). We frequently photographed active individuals using the slow burst mode of cameras, which allows to capture variation in face position and expression while avoiding identical frames. The multiple frames obtained when using the slow burst mode are hereafter referred to as a 'burst-mode series'. Images were then manually oriented to align pupils horizontally and cropped to generate square portraits centered on the nose and excluding the ears.

For training the DNN for both face identification and face verification, we used a learning dataset of 12.2 k pictures representing all individuals of the mandrill face database >1 yr, plus infants ≤1 yr not belonging to the 80 studied infants. Based on preliminary tests described previously (*Charpentier et al., 2020*), we included in the learning set all pictures displaying a face in frontal view (approx. <30°) and with less than 50% of occlusion, of whatever quality provided that field assistants individually recognize the animal. Because the face of a given individual varies considerably between its different age classes (infant, juvenile, adolescent -only for males-, subadult -only for males- and adult), we used 'ind-age' classes rather than individuals for the identification task, that is, we treated two ind-age classes representing the same individual as distinct and independent classes. The final learning dataset contained 168 ind-age classes. This learning set was split further into a training set and a validation set. The validation set was used to parameterize the model for the face identification task. It contained two images of each class. For a correct evaluation of training performances, we ensured that none of the validation image was from a burst-mode series that also contained images present in the training set. The training set contained all other images of the learning set. Because we were able to reach high performances despite a large imbalance between classes in the training sets, we did not attempt to correct for this imbalance.

The test dataset includes images of infants (≤1 yr) from the study natural population. To maximize the quality of dissimilarity measurements, we selected only the best-quality pictures: face in frontal view, sharp image, neutral light, occlusion limited to below the mouth, neutral facial expression, no shadow or lighting spot, single image of a burst-mode series. Eventually, we analyzed 204 images

representing 80 different infants (mean ± SD = 2.55±1.79 pictures per infant). For the sake of clarity, we alternatively use expressions like 'facial resemblance' (facial similarities) in the main text or 'facial distance' (facial dissimilarities). In other words, pairs of infants that look like each other present, on average, low facial distances.

## Face identification

We trained a DNN to identify ind-age classes as a goal to learn a deep representation of mandrill faces. We used the popular VGG-Face (*Parkhi et al., 2015*) as a starting point and retrained this network with mandrill portraits. This procedure, called 'transfer learning', allows to reach high model performance even with relatively small datasets (*Yosinski et al., 2014*). VGG-face is a VGG16 that previously learned to recognize 2.6 k different humans from a total of 2.6 M portrait pictures. We replaced the last softmax classification layer of VGG-face by a new layer of 168 dimensions (which corresponds to the number of classes in the new mandrill identification task). We included two dropout layers (with 50% dropout probability), one after each fully connected layer, to limit the risk of overfitting. We trained the network using a stochastic gradient descent with momentum optimizer, with initial learning rate of $10^{-5}$ for all convolutional layers except for the two fully connected layers and the classification layer, for which the learning rate was set to $10^{-3}$. The learning rate decreased by a factor 10 every 5 epochs. Learning continued until the validation loss did not decrease further after three consecutive epochs. In order to match the input size of VGG-face, mandrill portraits were downsized to 224×224 pixels ×3 colors prior to analyses. We set the batch size to 32. We limited overfitting further by using 'data augmentation': for each iteration, images were shifted horizontally and vertically (by a number of pixels randomly selected within the range [–40 40]), rotated (range of degree: [–20 20]) and scaled (range of factor: [–0.7 1.2]). The entire training procedure was repeated 10 times. Eventually, after approximately 15 epochs, VGG-Mandrill was able to identify mandrill faces individually with a maximal accuracy of 93.42% (sem:±0.13), and high generalization capacity (i.e. limited overfitting; *Appendix 1—figure 5*).

## Face verification

We retrained the DNN using the full learning set to maximize the number of images (i.e. with both the training and validation sets). VGG-Mandrill was then used to extract deep feature activation vectors, a compact (4,096-dimensional) and informative representation of a mandrill face. Previous experiments revealed that face verification with mandrills was the most performant when using activation vectors from the first fully connected layer (after RELU nonlinear transformation), compared to using vectors from the second, or from both fully connected layers (*Charpentier et al., 2020*). The distance between feature activation vectors predicts the resemblance between images (*Zhang et al., 2018*). In this study, we used a $\chi^2$ distance calculated with normalized features (for details on the normalization step, see *Charpentier et al., 2020*).

Last, we used a linear support vector machine (SVM) to learn a distance metric, with the goal to find the feature weights that optimize face verification. We randomly selected 15 k pairs of images representing different individuals and 15 k pairs representing same individuals, and for each pair we calculated the $\chi^2$ difference (as in *Parkhi et al., 2015*). We then ran the SVM with the $\chi^2$ differences as explanatory variables and 0 (different-individual pairs) or 1 (same-individual pairs) as a response variable. The SVM outputs the accuracy of the face verification task as well as the weight of each feature. We found that the classifier could verify mandrill identity from their face with an accuracy of 86.87% (sem: ±0.01). Weights were eventually used to calculate a weighted $\chi^2$ distance between every pairs of images in the test set. Pairwise distances were averaged for every different pairs of individual (N=3,160 pairs of infants represented by a total of 20,421 pairs of pictures) to provide our final estimates of facial distance across pairs of infants (mean ± SD: 6.46±6.99 pairs of pictures for each pair of infants).

## Behavioral observations

Trained observers, blind to the study questions, perform daily behavioral observations on all individually-recognized mandrills using either ad libitum data or 5 min focal sampling. During each focal observation, we record social interactions including time spent grooming or receiving grooming from all focal's groupmates. During these focal observations, we also record scans on one to three

occasions to measure spatial associations between the focal animal and all groupmates that are located less than five meters away (termed 'scan'). Since 2012, we obtained 9,305 scans from 48 infants and their 30 mothers during their offspring's first year of life (mean ± SD: 124.1±178.3 scans per individual). We restricted our analyses on association rates to infant-infant (169±77 scans per pair), mother-infant (174±80) and mother-mother (174±91) dyads with at least 5 scans (and at least one scan for either partner) and when both individuals were alive and infants aged ≤1 yr. For each association rate, we matched the average facial distance of the corresponding pair of infants aged ≤1 yr. For mother-mother dyads, we also retrieved the frequency of grooming exchanged, recorded during their offspring's first year of life and restricted our analysis to those dyads observed during at least 30 minutes (N=310 dyads; mean time of observation per dyad ± SD: 8.98±6.67 hours).

Behavioral observations further allowed to examine patterns of two conspicuous mother-infant behaviors, "follows" and "approaches" recorded as bouts during focal samplings of infants and their mothers. Finally, we reconstructed both male and female dominance hierarchies, using the outcomes of approach-avoidance interactions obtained during ad libitum and focal samplings and calculated using normalized David's score (as per *Charpentier et al., 2020*). We divided adult females into three classes of rank of similar size across the entire study period (2012–2020; high-ranking, medium-ranking, low-ranking). For males, we used monthly rank, distinguishing the alpha male from all other subordinates. When the alpha position was highly disputed for a given month, we considered the male's hierarchy as unclear (see also below).

## Matrilineal identity and genetic relatedness

Maternity was known for all 80 studied infants thanks to daily monitoring. Matrilineal identity was determined for all of them, including those 48 infants and associated 30 mothers used in the proximity analyses, thanks to detailed data on births and pedigree information available since the beginning of the project (*Charpentier et al., 2020*) and at CIRMF (*Charpentier et al., 2005*). In particular, individuals belong to the same matriline when they share a common mother, maternal grand-mother, great-grand mother and so-on with the exception of those females born at CIRMF that belonged to the same matriline but were released on two different occasions. In these cases, major social disruptions involved changes in female's dominance hierarchy, with females released in 2006 being automatically lower in rank than all other females released in 2002, whatever the initial rank they achieved. In these cases, females released in 2006 formed a new matriline, independent from their matrilineal origin at CIRMF.

Genetic relatedness was determined using pedigree data obtained from paternity analyses performed on most adult mandrills and on a subset of immature individuals (see for details *Charpentier et al., 2020*). We determined paternity for 32 of the 80 portrayed infants. These 32 infants were randomly sampled across all studied matrilines. For the proximity analyses, we restricted our data set to those infant dyads (and associated mother-infant and mother-mother dyads) for whom the mothers' pedigree was known to the previous generation (maternal grand-father and grand-mother known) in order to obtain good estimates of mother-mother relatedness. Relatedness across the 285 different pairs of mothers vary from 0.01 to 0.66 (mean ± SD: 0.13±0.12).

## Statistical analyses

All data were analyzed using R v. 3.6.1. using the package spaMM.

We first fitted a General Linear Mixed Model (LMM) to study variation in the averaged facial distance among pairs of 80 infants aged ≤1 yr as a function of the averaged distance in ages across all pairs of pictures collected on each studied pair of infants, their sex (class variable with three modalities: "FF", "FM", "MM"), whether or not they share a common mother (class variable with two modalities: "same mother", "different mother"), and whether or not they either share a common father and whether or not they were conceived during the tenure of the same alpha male (class variable with four modalities: "same father", "different father", "same alpha", "different alpha"). We excluded those infants (and the corresponding pairs) conceived during a month when the alpha position was disputed and unclear, resulting in 2,556 pairs of infants represented by 16,755 pairs of pictures (mean ± SD: 6.56±6.86 pairs of pictures for each pair of infants). We included an autocorrelated random effect with distinct levels for each pair, to represent correlations between pairs of facial distance values involving a same infant in both distances, with a single correlation value for all

such pairs. Its correlation is equal to 1 between permuted pairs (which are thus affected by the same value of the random effect), equal to a correlation value $\rho$ (fitted to the data) between pairs sharing one individual, and to 0 for non-overlapping pairs. The case where $\rho$ equals 0.5 represents the case of two additive individual random effects, each with the same variance. The fitted correlation model is therefore more general than this particular case. It is formally identical to a random effect model for a "diallel" experiment as considered in quantitative genetics, where the same individual may be the "first" or the "second" member of a mating pair across different mating pairs, and more specifically assuming that in each case it would express the same effect on its offspring (implying that there is a single variance parameter to be fitted). The present random effect structure accounts for the analogous symmetry of effects on facial distance. The case where $\rho$ =0.5 represents additive random effects from each individual (analogous to "general combining abilities" in quantitative genetics; *Lynch and Walsh, 1998*) and deviations from $\rho$ =0.5 additionally represent non-additive effects ("specific combining abilities"). We fitted a heteroscedastic residual variance with prior weights defined as the total number of pairs of pictures collected on each dyad of infants (giving more weight to those pairs with more numerous pictures). We standardized the averaged distance in ages to allow comparisons with other estimates.

We then fitted three Generalized Linear Mixed Models (GLMM) with negative binomial response family and log link to assess predictors of the frequency of spatial association among dyads of (i) infants ('infant-infant'); (ii) infants and other mothers ("mother-infant"); and (iii) mothers having infants in the social group ("mother-mother"). The response is the number of observed spatial associations. We considered only pairs of infants (and associated mothers) born into the same cohort (conceived during the same reproductive season). Each fitted model includes an offset (the logarithm of the total number of scans recorded on both individuals of the focal pairs), six fixed effects, a random effect term representing the effect of two additive individual random effects under the constraints of symmetry, as for "general combining ability" in a "diallel" experiment, in models of symmetrical interactions (mother-mother and infant-infant), and standard individual-level random effects for mothers and for infants in the mother-infant model. The six fixed effects are whether the infants belonged to the same matriline (class variable with two modalities: "same matriline", "different matriline"), relatedness between mothers (continuous variable), mothers' rank difference (continuous variable with three values: "0" for no rank difference, "1" for one rank difference, for example, between high-ranking and medium-ranking mothers; "2" for two rank differences, for example, between high-ranking and low-ranking mothers), infants' age difference when behaviors (not photographs) were recorded (continuous variable in days and <365), infants' sex (class variable with three modalities: "FF", "FM", "MM"), and the residuals of infant facial distance. Indeed, to assess the effect of infant facial distance unconfounded by the effect of infant age differences (at the time of pictures' collection), we used the residuals of such distance obtained from the fit of an LMM as performed above and that included as predictor the age difference (averaged distance in ages across all pairs of pictures), and an autocorrelated random effect with the same "diallel" correlation structure as above (N=3,160 pairs of infants, represented by 20,421 pairs of pictures).

Both matrilineal origin and females' relatedness were independent from facial distance values meaning that highly resembling infants were not more likely to be found within the same matriline than lowly resembling infants nor were they the offspring of highly related females (e.g., "infant-infant" data set: Pearson correlation between facial distance and mothers' relatedness: *r*=0.008, *P*=0.89; facial distance within same vs. different matriline, mean ± SD: 13.03±2.12, N=56 pairs of infants vs. 13.21±1.90, N=226 pairs of infants; and see Appendix 1).

Finally, we ran a last similar GLMM as above to assess the same predictors of grooming frequency among mothers (calculated as the frequency of grooming exchanged between each dyad of mothers).

We standardized all continuous predictors to allow comparisons of the estimates and we preliminary checked that no significant first-order interaction occurred among all these explanatory variables (not shown) and kept full models as final models. In all these three GLMM, we further checked for possible hazardous multicollinearities between continuous predictors by calculating variances of inflation (VIF <2). For the two significant negative effects of facial-distance residuals, we excluded individuals one at a time and checked that there was no apparent error in the data for individuals most supporting the negative relationships. To validate the models, we finally verified that the magnitude of standardized residuals is independent of the fitted values.

## Acknowledgements

We are grateful to past and present field assistants of the Mandrillus Project who collect daily behavioral data on the study population, to the Wildlife Reserves of Singapore and the Zoo of Granby and to the Primatological Centre at CIRMF (Gabon) for providing pictures of their mandrills, to the SODEPAL-COMILOG society (ERAMET group) for their long-term logistical support and contribution to the Mandrillus Project. This study was funded by several grants that allowed long-term collection of data: Deutsche Forschungsgemeinschaft (DFG, KA 1082-20-1) to PMK and MJEC, SEEG Lékédi (INEE-CNRS) to MJEC, and Agence Nationale de la Recherche to MJEC (ANR SLEEP 17-CE02-0002) and to JPR (ANR-20-CE02-0005-01). This study was approved by an authorization from the CENAREST institute (permit number: AR017/22/MESRSTTCA/CENAREST/CG/CST/CSAR). This is a Project Mandrillus publication number 28 and ISEM 2022-252 SUD.

## Additional information

### Funding

| Funder | Grant reference number | Author |
|---|---|---|
| Deutsche Forschungsgemeinschaft | KA 1082-20-1 | Marie JE Charpentier Peter M Kappeler |
| Agence Nationale de la Recherche | SLEEP 17-CE02-0002 | Marie JE Charpentier |
| Agence Nationale de la Recherche | ANR-20-CE02-0005-01 | Julien P Renoult |
| SEEG | Lékédi | Marie JE Charpentier |

The funders had no role in study design, data collection and interpretation, or the decision to submit the work for publication.

### Author contributions

Marie JE Charpentier, Conceptualization, Data curation, Formal analysis, Funding acquisition, Validation, Investigation, Methodology, Writing – original draft, Project administration, Writing – review and editing; Clémence Poirotte, Conceptualization, Formal analysis, Validation, Investigation, Methodology, Writing – review and editing; Berta Roura-Torres, Paul Amblard-Rambert, Data curation, Investigation; Eric Willaume, Project administration; Peter M Kappeler, Funding acquisition, Validation, Writing – review and editing; François Rousset, Conceptualization, Formal analysis, Validation, Investigation, Methodology, Writing – original draft, Writing – review and editing; Julien P Renoult, Conceptualization, Data curation, Formal analysis, Funding acquisition, Validation, Investigation, Methodology, Writing – original draft, Writing – review and editing

### Author ORCIDs

Marie JE Charpentier ⬩ http://orcid.org/0000-0001-6530-5874

### Decision letter and Author response

Decision letter https://doi.org/10.7554/eLife.79417.sa1
Author response https://doi.org/10.7554/eLife.79417.sa2

## Additional files

### Supplementary files

• MDAR checklist

### Data availability

All data needed to evaluate the conclusions are provided in the article and/or as a Supplementary Information and have been deposited on DRYAD (see: https://doi.org/10.5061/dryad.dbrv15f3m). The Mandrillus Face database is composed of >40,000 pictures and is thus not downloadable but the

pictures of infants used in this particular study are available on Dryad, accessible here: https://doi.org/10.5061/dryad.gtht76hqb.

The following datasets were generated:

| Author(s) | Year | Dataset title | Dataset URL | Database and Identifier |
|---|---|---|---|---|
| Charpentier MJ, Poirotte C, Roura-Torres B, Amblard-Rambert P, Willaume E, Kappeler P, Rousset F, Renoult J | 2022 | Mandrill mothers associate with infants who look like their own offspring using phenotype matching | https://doi.org/10.5061/dryad.dbrv15f3m | Dryad Digital Repository, 10.5061/dryad.dbrv15f3m |
| Charpentier MJ, Poirotte C, Roura-Torres B, Amblard-Rambert P, Willaume E, Kappeler P, Rousset F, Renoult J | 2022 | Mandrillus face database: Portrait pictures of the population of wild mandrills from Bakoumba (Gabon) | https://doi.org/10.5061/dryad.gtht76hqb | Dryad Digital Repository, 10.5061/dryad.gtht76hqb |

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

# Appendix 1

## Maternal and paternal half-sibs are equally related

Both paternity and maternity were unambiguously genetically determined for a subset of 81 individuals from the study group (see for details *Charpentier et al., 2020*). On these 81 individuals, 72 dyads were maternal half-sibs (with known fathers) and 371 were paternal half-sibs (with known mothers). We excluded 17 full-sib dyads. Pairwise genetic relatedness was calculated from the pedigree for these maternal and paternal half-sibs using ENDOG version 4.8 (*Gutiérrez and Goyache, 2005*). The average genetic relatedness of maternal half-sibs was: 0.286±0.044 (mean ± SD) while the average genetic relatedness of paternal half-sibs was: 0.295±0.035, suggesting that both kin categories are equally genetically related, as expected.

## Possible evolutionary functions of paternal kin discrimination in mandrills and associated proximate mechanisms

**Appendix 1—table 1.** The first table (**a**) presents possible mechanisms of paternal kin recognition to explain increased nepotism among paternal half-sibs in mandrills (*Charpentier et al., 2007*; *Charpentier et al., 2020*), and their associated empirical evidence (or lack thereof) with respect to an *increased* facial resemblance among paternal half-sibs; the second table (**b**) lists the empirical evidence (or lack thereof) for different possible evolutionary functions.

a.

| Possible mechanism(s) | Empirical data in mandrills | Likelihood in paternal half-sib mandrills |
|---|---|---|
| Self-phenotype matching | Absence of any physical substrate (mirror) to allow facial self-recognition | **Low**. In addition, facial similarity as a by-product of selection on other self-evaluable phenotypic traits (e.g. odours) would not explain *increased* facial resemblance among paternal half-sibs |
| Phenotype matching with father as a template | Adult male mandrills do not resemble their offspring at any age because of pronounced sexual dimorphism (see *Appendix 1—figure 1*) | **Low**. |
| **Familiarity mediated by fathers:** Males know with whom they reproduced and associate either with the potential mothers or their offspring (or both) following births favoring secondary association among paternal half-sibs | Male mandrills are not responsible for establishing spatial associations with either females or infants (MJEC and BRT, Pers. Obs.) -Half of the male mandrills present during a reproductive season are absent from the group the next birth season (this study) | **Low**. In addition, these two mechanisms alone fail to explain *increased* facial resemblance among paternal half-sibs that *necessarily* involves some forms of phenotype matching |
| **Familiarity mediated by mothers:** Females know with whom they reproduced and associate either with the father, the other females that also reproduced with the same father or their offspring (or a combination of all three) following births, favoring secondary association among paternal half-sibs | -Mother-mother association and grooming relationships do not depend on facial similarity among offspring (this study) -Possible memory and field constraints as females would need to remember all copulation events that occurred with other females in a dense habitat | |

*Appendix 1—table 1 Continued on next page*

*Appendix 1—table 1 Continued*

**a.**

| Possible mechanism(s) | Empirical data in mandrills | Likelihood in paternal half-sib mandrills |
|---|---|---|
| **A mix between familiarity and phenotype-matching+father mediation:** Males know with certainty at least one paternity (e.g. thanks to patterns of copulations) and associate either with the mothers or their offspring (or both) following births when these offspring resemble more to their own (certain) offspring, favoring secondary association among paternal half-sibs | -Male mandrills are not responsible for establishing spatial associations with either females or infants (MJEC and BRT, Pers. Obs.) -Half of the male mandrills present during a reproductive season are absent from the group the next birth season (this study) | **Low.** This mechanism may explain *increased* facial resemblance among paternal half-sibs but is not parsimonious given males' patterns of residency and the fact that males are not responsible for establishing proximities with infants or their mothers. In addition, it requires at least one event of paternity certainty. This mechanism also fails to explain why mothers associate more with strange but resembling infants but not with their mothers if fathers mediate these associations |
| **A mix between familiarity and phenotype-matching+mother mediation:** Mothers associate either with the mothers or their offspring (or both) following births when these offspring resemble their own (certain) offspring more, favoring secondary association among paternal half-sibs | Mothers associate more with strange infants that resemble their own offspring more, possibly favoring secondary association among paternal half-sibs (this study) | **High.** This mechanism may explain *increased* facial resemblance among paternal half-sibs and their long-term nepotism. Mother mediation could have also been selected to favor paternal care and offspring protection against infanticide (patterns of mother-infant or infant-infant association would be by-product of association to a father), but it would not explain alone the *increased* facial resemblance among paternal half-sibs |

**b.**

| Evolutionary functions | Empirical data in mandrills | Likelihood in mandrills |
|---|---|---|
| Nepotism | -Paternal half-sibs are numerous in the group (>2 times more numerous than maternal half-sibs) because of high male reproductive skew (*Charpentier et al., 2020*; *Charpentier et al., 2005*) -Nepotism occur among paternal half-sibs in juvenile and adult female mandrills (*Charpentier et al., 2020*; *Charpentier et al., 2007*) | **High.** Potentially elevated fitness benefits to recognize and favor paternal half-sibs and strong empirical support |
| Inbreeding avoidance | -Alpha males' tenure (<15 months) and males' length of stay in the group (<23 months for males aged ≥10 yrs) are restricted -Sex-biased dispersal: most males emigrate before being reproductive (<7 yrs; males start reproducing around 10 yrs) | **Low.** Father-daughter reproduction is highly unlikely; inbreeding among paternal half-sibs may occasionally occur if males reproduce with their sisters before emigrating; mothers should avoid rather than favor association with highly resembling infants; other mechanisms have probably evolved, such as sex-biased dispersal |
| Paternal care and offspring protection against infanticide | -Social relationships between adult males and infants are highly limited (e.g. absence of affiliation) although, in captivity, fathers are spatially closer to their own offspring than to unrelated juveniles (*Charpentier et al., 2007*) -There is indirect evidence of infanticide in mandrills -Only 54.5% of alpha males and 44.7% of subordinates are present during the birth season following the reproductive season they experienced (based on 69 male.years) -When present, males stay less than a year in their offspring's group (9.5 months on average for alpha males; 5.5 months for subordinates; based on 33 male.years) | **Medium.** Patterns of male residency do not offer strong support for this hypothesis, but paternal care may occur early in life in the form of increased spatial association or support during agonistic interactions; offspring protection against infanticide may also occur |

## Within cohort relatedness among mothers is not higher than among cohorts

In the matrilineal society of mandrills, maternal kin females show highly differentiated social relationships (*Charpentier et al., 2020*; *Charpentier et al., 2007*) (and for further information on the biology and sociality of mandrills, see *Dezeure et al., 2022*; *Brockmeyer et al., 2015*). It is tempting to seek an explanation for the correlations between mother-infant associations and facial similarity in the fact that both variables may be correlated to a third one, mother-mother relatedness. However, although such correlations exist, they can be rejected as an explanation for our results on several grounds. First, the effects of facial similarity and relatedness were considered simultaneously in the statistical analyses, with standard inference methods that should not infer an effect of a predictor variable X1 (here facial similarity) correlated to another predictor X2 (here relatedness) if only X2 has an effect on the response variable (i.e if there are causal paths through relatedness to response and through relatedness to facial similarity but no path through facial similarity to response, disjoint from the one through relatedness). Nevertheless, the effect of relatedness might itself be poorly inferred if average relatedness levels varied strongly between years (while it is only variation in relatedness within years that should explain choice of alternative social partners within years). On the contrary, mean within-year mother-mother relatedness did not vary among years. To support this claim, we have measured the average relatedness of females (with both parents known; N=37 females) that reproduced a given year and compared it to these females that did not reproduce that year (7 yrs of data, N=2,830 dyads; females that did reproduce (mean relatedness ± sd): 0.148±0.022; females that did not reproduce: 0.135±0.041; Wilcoxon Signed-rank test: p=0.81). Mother-mother relatedness is thus variable within years, in particular because 30–40% of infants born into a given cohort are not paternal half-sibs (*Charpentier et al., 2007*; *Charpentier et al., 2017*).

Further, one might also speculate about kin selection effects favoring associations if within-cohort mother-mother relatedness is higher that among-cohort relatedness. However, beyond the fact that it would still not explain the demonstrated distinct effect of facial similarity, no such variation in relatedness is detectable. These females that reproduced a given year were not more related to each other (0.148±0.022) than females that reproduced the other years (0.145±0.003; Wilcoxon Signed-rank test: p=1). In addition, and for the subset of infants for whom the father was known (N=15), the average relatedness among mothers of paternal half-sibs (N=22 dyads) was: 0.178±0.110 while the average relatedness among mothers of offsprings sired by different fathers (N=80 dyads) was: 0.187±0.132 (Student t-test: t=0.32, p=0.75).

Consequently, mother-mother relatedness did not influence females' probability to reproduce a given year and thus cannot explain our main results. Similar remarks can be made about matrilineal origin instead of relatedness. A linear model studying the percentage of females from any given matriline (8 different matrilines in the study group) that reproduced a given year (N=50 matriline. year) as a function of the year of reproduction and of the identity of the matriline detected no effect of these two variables (year: *F*=0.73, p=0.63; matriline identity: *F*=0.52, p=0.82).

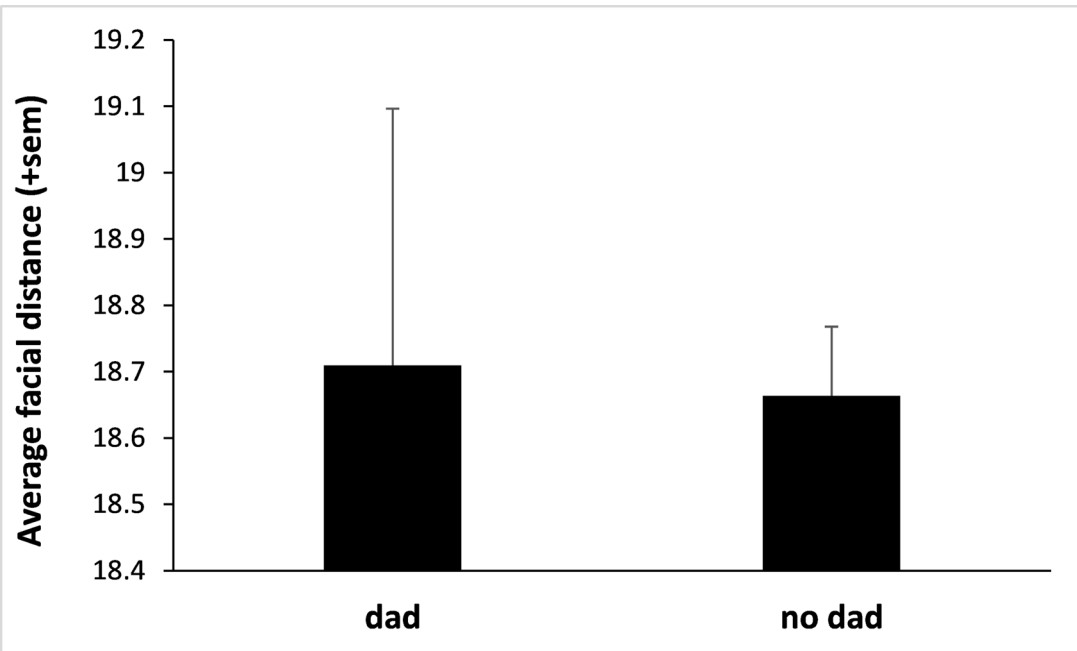

**Appendix 1—figure 1.** Fathers do not resemble their own offspring. We used deep neural network analyses to estimate father-offspring resemblance (using the same methodology as for infant-infant resemblance; see main text) and did not find evidence that infants resemble their father. We considered the 32 infants for which the father was known (sired by 12 different fathers) and compared their facial distances across dyads of father-offspring and non-father-infant (N=47,724 pairs of pictures from a total of 590 male and infant portraits) using a LMM (response variable: facial distance; explanatory variables: infant's sex and whether the male was the father or not). We considered the identity of the adult male and the identity of the infant as two random variables. We fitted a heteroscedastic residual variance with prior weights defined as the total number of pairs of pictures collected on each dyad of infants (giving more weight to those pairs with more numerous pictures). We found that infants do not resemble their father ('dad'; figure above) more than any random male ('no dad'; N=352 pairs; p=0.39 by likelihood ratio test; there is also no effect of the sex of the infant: p=0.74). The highly pronounced sexual dimorphism and morphological differences between immatures and adults in this species likely explain this result.

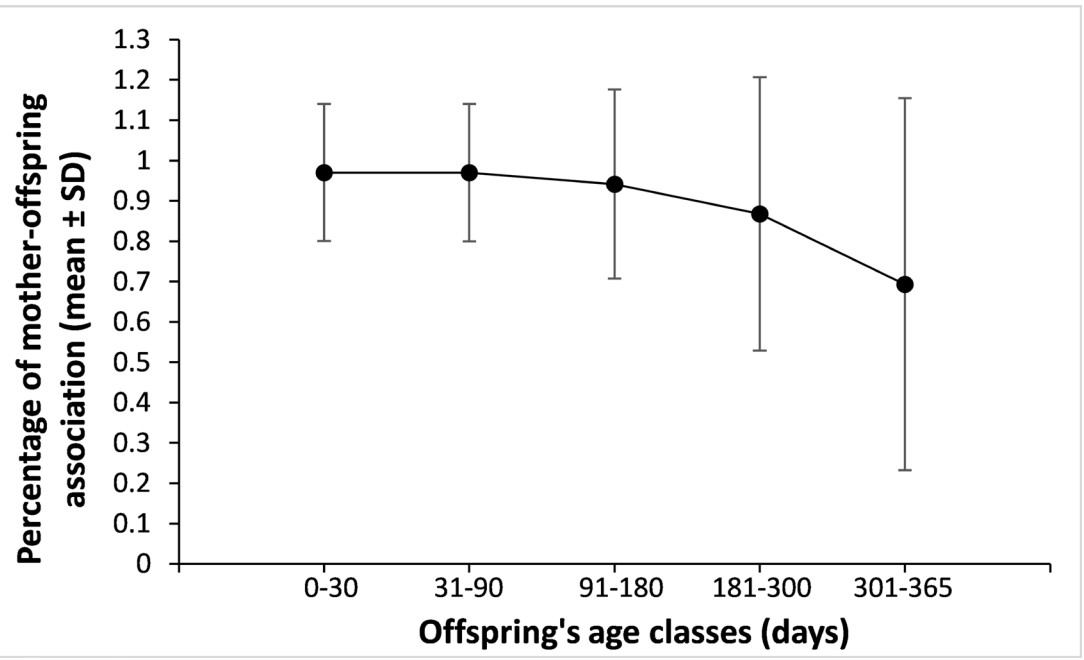

**Appendix 1—figure 2.** Mother-offspring spatial association across offspring's age. The figure above represents the percentage of scans during which the mother was associated (0–5 m) to her offspring averaged across different offspring's age classes (in days). The figure was obtained from a total of 11,926 scans performed on 217 different infants aged 0–1 yr (for 11,003 scans, the mother was 0–5 m away from her offspring). Sample sizes are as follow: 0–30 days: 2,255 scans; 31–90 days: 3,549 scans; 91–180 days: 3,148 scans; 181–300 days: 1,983 scans; 301–365 days: 991 scans. Although, this pervasive mother-offspring association should, intuitively, translate into high mother-mother association (that we do not observe with respect to infant-infant facial resemblance), the variance observed is high, suggesting strong variation in association patterns across mother-offspring dyads. In addition, mothers may be located 5 m away or less from their offspring and from other similar-looking infants without being less than 5 m away from these infants' mothers.

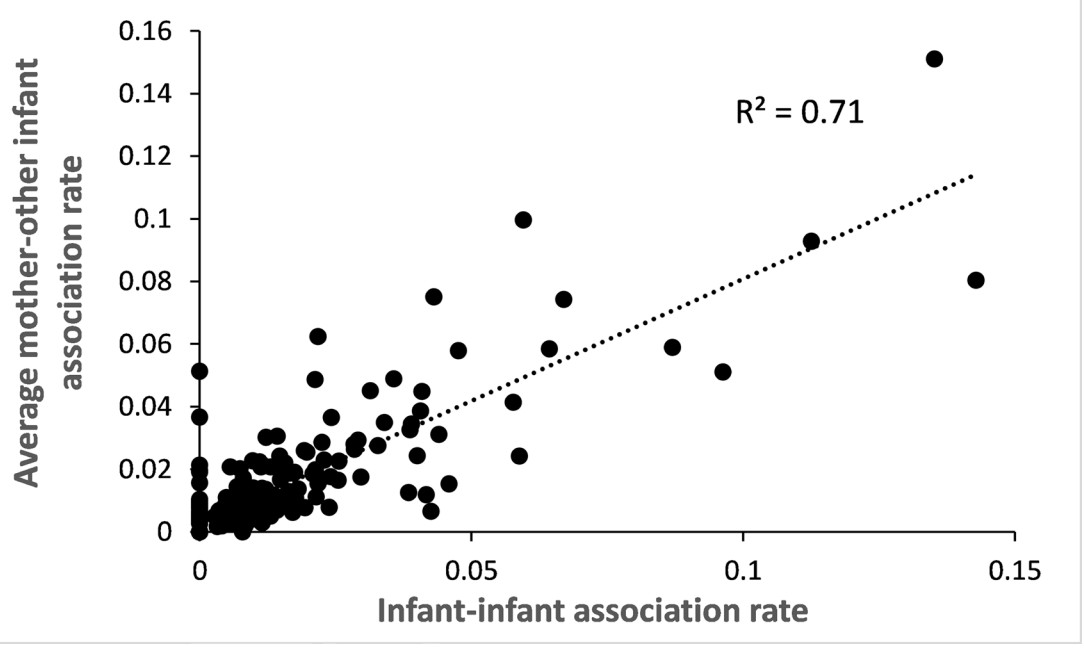

**Appendix 1—figure 3.** Correlation between infant-infant association rate and mother-other infant association rate. For the 282 study pairs of infants, we calculated the raw rate of association per dyad of infants (total number
*Appendix 1—figure 3 continued on next page*

*Appendix 1—figure 3 continued*

of scans where both infants were in proximity divided by the total number of scans performed on the two members of the dyad) and plotted it against the average raw rate of association of each mother with the other infant. We found that both association rates are highly positively correlated (Pearson correlation test: N=282, *r*=0.85, p<0.0001).

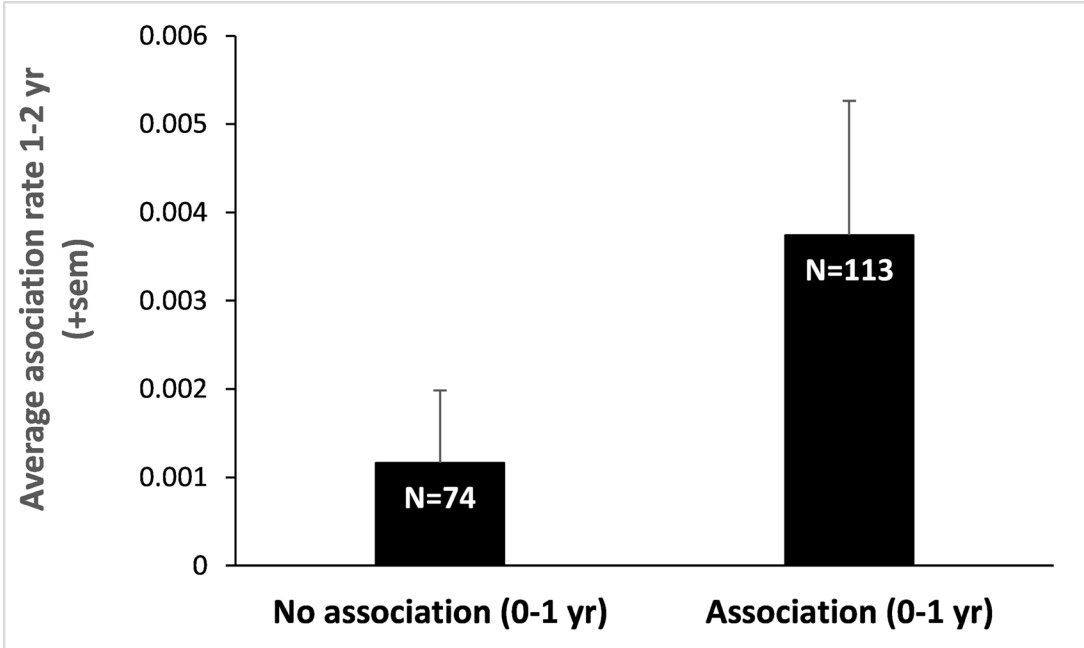

**Appendix 1—figure 4.** Infant-infant association rates a year later. We split our initial data set on infant-infant associations between those dyads that were associated early in life (0–1 year) and those that were not. We then retrieved the association rates (number of scans where both individuals were recorded in proximity divided by the total number of scans performed on them both) of these infants (those that survived and for whom we had detailed data on spatial association), a year later (when they were aged 1–2 years). Infants that did associate at 0–1 year also associate, on average, three times more at 1–2 years than those that did not. Sample sizes are provided within bars and represent the number of dyads of juveniles aged 1–2 years. This result, although preliminary, suggests that early association during infancy may pervade later in life.

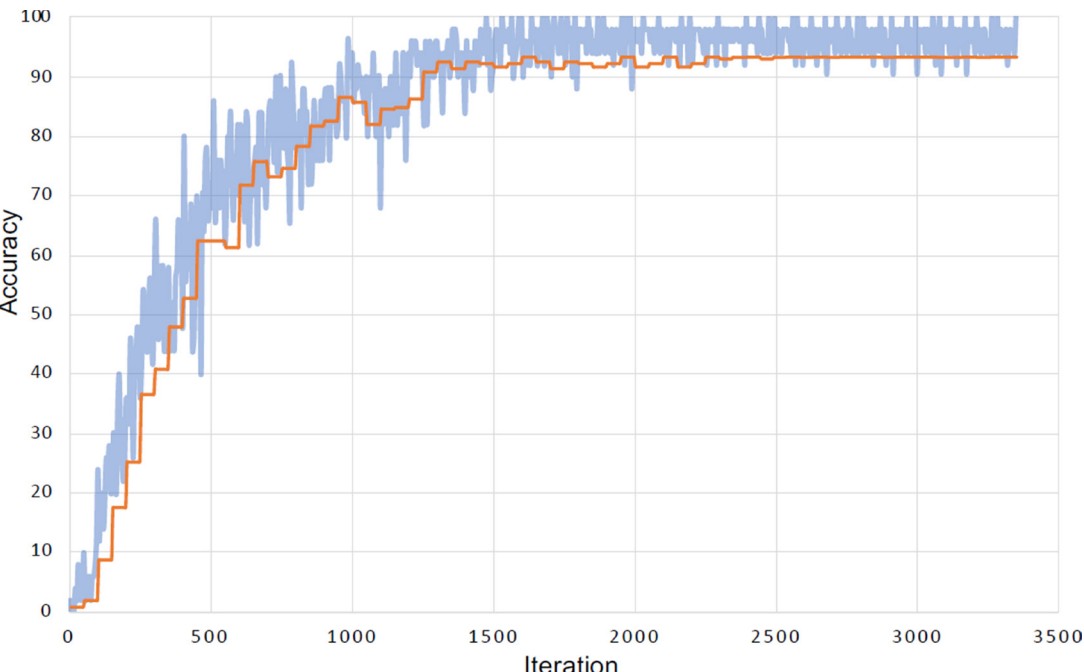

**Appendix 1—figure 5.** Learning curve for the face identification task. Evolution of accuracy for the training set (in blue) and validation set (in orange) during the run that yielded the highest accuracy (93.42%). The small difference between the training accuracy and the validation accuracy indicates limited or no effect of overfitting.

