## [Editor Report]

This study shows that, 60 years after the development of kin selection theory, new implications are still being uncovered. The authors report results of a long-term field study of a mandrill population in the forests of Gabon. Facial-pattern analyses and accompanying theoretical work support the hypothesis that females "socially engineer" relationships between their offspring and other offspring, based on facial resemblance. Via this mechanism, mothers appear to promote associations with individuals that are more likely to treat them as relatives, increasing the likelihood of future benefits from cooperative interactions. The authors suggest that their discovery could explain cooperative behaviours among non-kin in other social species, including humans.

---

## [Decision Letter]

**Decision letter after peer review:**

Thank you for submitting your article "Primate mothers promote proximity between their offspring and infants who look like them" for consideration by *eLife*. Your article has been reviewed by three peer reviewers, one of whom is a member of our Board of Reviewing Editors, and the evaluation has been overseen Christian Rutz as the Senior Editor. The following individual involved in the review of your submission has agreed to reveal their identity: James Higham (Reviewer #3).

The reviewers have discussed their reviews with one another, and the Reviewing Editor has drafted this decision letter to help you prepare a revised submission. The reviewers were unanimous in identifying the need for more data to support the conclusions as stated. The decision to invite resubmission was marginal -- please carefully revise your article, to address the points raised below. We are planning to send out the revised article for re-evaluation.

Essential revisions:

1) The manuscript should be revised to more accurately report the level of support for the hypothesis of phenotype-matching by mothers to promote kin-selected cooperation towards offspring. The evidence in support of the single hypothesis is currently overstated. A more balanced article would be considerably toned down, with appropriate notes of caution about the evidence for the proposed mechanism, critical tests, and alternative explanations. What is the evidence for potential benefits from sib interactions?

2) A revision should help readers assess the likelihood of alternative hypotheses: paternal sibs look alike and associate more with each other than expected, but to what extent can we attribute maternal behaviour to explain this?

Note: Please note that *eLife* has adopted the STRANGE framework, to help improve reporting standards and reproducibility in animal behaviour research. In your revision, please consider scope for sampling biases and potential limitations to the generalisability of your findings:

https://reviewer.elifesciences.org/author-guide/journal-policies

https://doi.org/10.1038/d41586-020-01751-5

*Reviewer #1 (Recommendations for the authors):*

The paper is generally clear and well-written.

Lines 102-103: Please drop unnecessary acronyms: MHS and PHS. Seems designed purely to save having to type and they are confusing as 'S' of MHS refers to sisters, but 'S' of PHS refers to sibs.

Line 115: Suggest a new paragraph to start from "In Charmentier…"

Lines 142-143: Perhaps a little more context here would be helpful about the study population. For example, perhaps important to note that the group is semi-captive in the main text.

Figure 1: My only significant difficulty is with interpreting Figure 2 -- the figure showing the main result. From the text from which it is referred to (lines 209 and 211), I'm looking for evidence of an association between females and offspring that look like their own offspring. None of the Y-axis categories seems to correspond to this group. The "variable of interest" is presented in pink as "facial distance". The legend was not helpful at all.

*Reviewer #2 (Recommendations for the authors):*

Suggestions to improve the paper.

1. The spatial proximity data are interesting but suggestive and preliminary. I would therefore recommend toning down the interpretation in places. For example, I think the subheading "The driving role of mothers" is too strong. You haven't shown this. Also, the second paragraph of the Conclusion seems to be arguing strongly that spatial proximity scores, as measured here, must convert into benefits in the longer term. That might be the case but not necessarily so; for example, there might be local competition effects.

2. I wasn't clear why you did not analyse any other data to try to test your hypothesis. In line 570 you state "Here, we exclusively studied association rates and no other social behavior such as grooming because scans were more numerous than any other behavioral records, and also because we reasoned that if mothers influence their offspring's social opportunities, this will mainly take the form of increased associations and not of an increased investment into potentially costly social behavior." I don't understand this reasoning. First, just because scans are more numerous is not a reason in itself to ignore other behavioural data. Were the other data too scant to be analysed? Did you analyse them, but not include them, because there was no pattern? A bit of transparency would be helpful here.

Second, it doesn't seem a good reason to not analyse something because you 'think' that's not the way second-order kin selection will work. For example, mothers might bring infants into proximity with each other, and they do the rest; or mothers might be more active in the process. Presumably, you don't know until you examine the data. So neither of these reasons for excluding information are good ones.

3. In the second paragraph of the conclusion you state that "a firm demonstration of the second condition [namely the occurrence of reciprocal interactions between the matched social partners] will necessitate years of detailed behavioural and demographic data and is beyond the scope of this paper"; this seems a bit odd -- like you are trying to convince the reader that you haven't done any other tests because they would all take years. Are there really no other tests that could be done in a shorter time frame, or with the data you have?

4. Finally, I wondered about other factors that might contribute to the enhanced facial similarity in PHS compared to MHS. If I have understood correctly, paternal half-sibs are offspring born to different mothers fathered by the same α, whereas maternal half-sibs would be born to the same mother with different fathers. This would seem to imply that PHS is clustered temporally together (within the tenure of the same α) whereas the MHS are necessarily spread out temporally (across two different α male tenures). Is that the case, and can you rule out temporal clumping as a factor influencing patterns of variation in the data?

*Reviewer #3 (Recommendations for the authors):*

I think that an appropriate reworking would present the manuscript differently, as a demonstration that paternal siblings look alike and that they associate. The authors could then go on to explore different possible explanations for this using their association data, make the case that maternal behavior is the most plausible (but not the only) explanation, and present their model of how such behavior could bring fitness benefits. In my view, such a paper would be publishable, but the current presentation is problematic.

---

## [Author Response]

Essential revisions:1) The manuscript should be revised to more accurately report the level of support for the hypothesis of phenotype-matching by mothers to promote kin-selected cooperation towards offspring. The evidence in support of the single hypothesis is currently overstated. A more balanced article would be considerably toned down, with appropriate notes of caution about the evidence for the proposed mechanism, critical tests, and alternative explanations.

We have now revised our manuscript to tone down the interpretation of our results and give more weight to alternative mechanisms involving fathers. For example, we have changed the title (now: “Mandrill mothers associate with infants who look like their own offspring using phenotype matching”) to report more accurately our main findings, and changed/added several sentences/paragraphs throughout the manuscript (see details in the responses to referees below).

What is the evidence for potential benefits from sib interactions?

In two previous publications (Charpentier et al. 2007; 2020), we have demonstrated, both in captivity and in the wild, that mandrills bias their social behavior preferentially towards close maternal and paternal kin compared to non-kin. In particular, male and female juvenile mandrills affiliate more with their adult paternal and maternal half-sisters than any other adult females (except their mother); adult females disproportionally groom and associate, but also fight, more with their maternal and paternal half-sisters than with non-relatives. Overall, maternal and paternal half-sisters constitute privileged social partners in their group (an information that we have now added on L90-92). Although the multiple fitness benefits of social integration are widespread throughout the animal kingdom (Snyder-Mackler et al. 2020), in our specific case these kin-driven benefits are not yet known. Estimating fitness benefits of social behavior would necessitate long-term data collection on fitness proxies such as lifetime reproductive success, that are not available yet. Our results are, however, consistent with a wealth of studies showing kin-biased social behaviors, and the predictions of kin selection theory.

2) A revision should help readers assess the likelihood of alternative hypotheses: paternal sibs look alike and associate more with each other than expected, but to what extent can we attribute maternal behaviour to explain this?

In the revised version of our manuscript, we now thoroughly discuss alternative mechanism involving fathers (L61-62; L245-253; L374-376). In addition, to address the extent to which maternal behavior contributes to infant-infant association, we have calculated the correlation between the raw rate of spatial association observed among pairs of infants as a function of the average rate of association recorded between each of these infant and the mother of the other infant. Appendix 1 – figure 3 highlights the clear relationship between the two variables: about 71% of the variance in infant-infant association is explained by the average association between each mother and the other infant (and *vice versa*). We have now included this information in the revised version of our manuscript and have discussed it (L201-207).

Reviewer #1 (Recommendations for the authors):The paper is generally clear and well-written.

Thank you for this positive comment.

Lines 102-103: Please drop unnecessary acronyms: MHS and PHS. Seems designed purely to save having to type and they are confusing as 'S' of MHS refers to sisters, but 'S' of PHS refers to sibs.

Changed as suggested (and throughout the manuscript, for clarity).

Line 115: Suggest a new paragraph to start from "In Charmentier…"

Done.

Lines 142-143: Perhaps a little more context here would be helpful about the study population. For example, perhaps important to note that the group is semi-captive in the main text.

This is not the case (see detailed explanation above; we have added “natural” to clarify this point; L140).

Figure 1: My only significant difficulty is with interpreting Figure 2 -- the figure showing the main result. From the text from which it is referred to (lines 209 and 211), I'm looking for evidence of an association between females and offspring that look like their own offspring. None of the Y-axis categories seems to correspond to this group. The "variable of interest" is presented in pink as "facial distance". The legend was not helpful at all.

Figure 2 represents the estimates of the models (performed using facial distances not resemblances). Consequently, a negative estimate (like the one found for facial distances) indicates a negative correlation between spatial association and that variable (facial distance): individuals associate more with low values of facial distances (high resemblances). We have added this information in an updated legend (“Note that a negative estimate (as for “facial distance”) indicates a negative correlation between spatial association and that variable (“facial distance”): individuals associate more with low values of “facial distance” (high resemblance)”).

Reviewer #2 (Recommendations for the authors):Suggestions to improve the paper.1. The spatial proximity data are interesting but suggestive and preliminary. I would therefore recommend toning down the interpretation in places. For example, I think the subheading "The driving role of mothers" is too strong. You haven't shown this. Also, the second paragraph of the Conclusion seems to be arguing strongly that spatial proximity scores, as measured here, must convert into benefits in the longer term. That might be the case but not necessarily so; for example, there might be local competition effects.

We fully agree with this comment. We have therefore changed the subheading “The driving role of mothers” to “Mothers and their offspring associate more with similar-looking other infants”. We have further toned down the interpretation of our findings throughout the text. For example, on lines 61-62; 245-253; 374-376.

2. I wasn't clear why you did not analyse any other data to try to test your hypothesis. In line 570 you state "Here, we exclusively studied association rates and no other social behavior such as grooming because scans were more numerous than any other behavioral records, and also because we reasoned that if mothers influence their offspring's social opportunities, this will mainly take the form of increased associations and not of an increased investment into potentially costly social behavior." I don't understand this reasoning. First, just because scans are more numerous is not a reason in itself to ignore other behavioural data. Were the other data too scant to be analysed? Did you analyse them, but not include them, because there was no pattern? A bit of transparency would be helpful here.

We agree with this comment too. We have now included similar models on grooming relationships between mothers as a function of their offspring’s facial distance. We show that mothers do not groom each other more when their offspring resemble each other more (as for spatial association). In addition, at these young ages, infants do not groom each other yet; we did not thus add a similar model for infant-infant grooming relationships. Finally, in the mother-infant model, only 6 females groomed other infants over 560 mother-infant dyads, precluding any reliable statistical analysis, and suggesting that, as we proposed, females do not invest into social relationships with other infants. As noted by this reviewer below, mothers thus seem to simply bring their offspring into proximity to resembling infants who possibly do the rest. We have now introduced and discussed this new analysis (L184-193).

Second, it doesn't seem a good reason to not analyse something because you 'think' that's not the way second-order kin selection will work. For example, mothers might bring infants into proximity with each other, and they do the rest; or mothers might be more active in the process. Presumably, you don't know until you examine the data. So neither of these reasons for excluding information are good ones.

We agree and have addressed this comment by including new analyses on grooming relationships and have further discussed these new results (L184-193; L531-534).

3. In the second paragraph of the conclusion you state that "a firm demonstration of the second condition [namely the occurrence of reciprocal interactions between the matched social partners] will necessitate years of detailed behavioural and demographic data and is beyond the scope of this paper"; this seems a bit odd -- like you are trying to convince the reader that you haven't done any other tests because they would all take years. Are there really no other tests that could be done in a shorter time frame, or with the data you have?

This sentence was indeed awkward (ideally, we would have liked to study whether those infants that associate more during infancy also stay associated once adults but we still do not have these longitudinal data). However, to address this comment, at least partially, we have looked at the association rates of the study pairs of infants, a year later (when they were aged 1-2 yrs). Compared to infants that did associate at 0-1 yr, those that did not, associate three times less on average at 1-2 yrs. Note that the sample size of juveniles aged 1-2 yrs decreased drastically because some of them died and until 2020, we were unable to physically recognize some of these former infants once weaned (sample sizes are provided within bars and represent the number of pairs of 1-2 yrs juveniles). This result, although preliminary, suggests that early association during infancy may pervade later in life. We have added this new figure in Appendix 1 – figure 4; and see also: (L389-390).

4. Finally, I wondered about other factors that might contribute to the enhanced facial similarity in PHS compared to MHS. If I have understood correctly, paternal half-sibs are offspring born to different mothers fathered by the same α, whereas maternal half-sibs would be born to the same mother with different fathers. This would seem to imply that PHS is clustered temporally together (within the tenure of the same α) whereas the MHS are necessarily spread out temporally (across two different α male tenures). Is that the case, and can you rule out temporal clumping as a factor influencing patterns of variation in the data?

This is a very good and accurate point: MHS are always at least a year apart while some PHS are, indeed, born in the same cohort, although not all of them as some males reproduced either a few consecutive years (generally no more than two) or a few years apart, due to various patterns of male migration found in this population. In our data set of 80 infants, for example, MHS are, on average, 3.5 yrs apart while PHS are 0.5 yrs apart. This is the reason why, in our analyses of spatial association, we only considered dyads of infants born into the same cohort, an information that we have now justified in the revised version of our manuscript (L596-597). Our main results, that mother and infants associate more with similarly looking other infants, are thus found on cohorts of individuals that are all similar in age. Temporal clumping thus cannot explain our results (although we agree that similarity in age impacts social relationships in addition to patrilineal and matrilineal kinship).

Reviewer #3 (Recommendations for the authors):I think that an appropriate reworking would present the manuscript differently, as a demonstration that paternal siblings look alike and that they associate. The authors could then go on to explore different possible explanations for this using their association data, make the case that maternal behavior is the most plausible (but not the only) explanation, and present their model of how such behavior could bring fitness benefits. In my view, such a paper would be publishable, but the current presentation is problematic.

For the sake of clarity, most of the alternative scenarios were initially addressed in a Supplementary Table (S2; now Appendix 1 – table). Perhaps that the information, somehow, got lost. We have therefore discussed more widely these alternative mechanisms in the revised version of our manuscript (e.g. L61-62; L245-253; L374-376).